# Research Advances on the Damage Mechanism of Skin Glycation and Related Inhibitors

**DOI:** 10.3390/nu14214588

**Published:** 2022-11-01

**Authors:** Wenge Zheng, Huijuan Li, Yuyo Go, Xi Hui (Felicia) Chan, Qing Huang, Jianxin Wu

**Affiliations:** 1Skin Health and Cosmetic Development & Evaluation Laboratory, China Pharmaceutical University, Nanjing 210009, China; 2Royal Victoria Hospital, BT12 6BA Belfast, Northern Ireland, UK

**Keywords:** skin glycation, advanced glycation end products (AGEs), anti-glycation, AGEs inhibitors

## Abstract

Our skin is an organ with the largest contact area between the human body and the external environment. Skin aging is affected directly by both endogenous factors and exogenous factors (e.g., UV exposure). Skin saccharification, a non-enzymatic reaction between proteins, e.g., dermal collagen and naturally occurring reducing sugars, is one of the basic root causes of endogenous skin aging. During the reaction, a series of complicated glycation products produced at different reaction stages and pathways are usually collectively referred to as advanced glycation end products (AGEs). AGEs cause cellular dysfunction through the modification of intracellular molecules and accumulate in tissues with aging. AGEs are also associated with a variety of age-related diseases, such as diabetes, cardiovascular disease, renal failure (uremia), and Alzheimer’s disease. AGEs accumulate in the skin with age and are amplified through exogenous factors, e.g., ultraviolet radiation, resulting in wrinkles, loss of elasticity, dull yellowing, and other skin problems. This article focuses on the damage mechanism of glucose and its glycation products on the skin by summarizing the biochemical characteristics, compositions, as well as processes of the production and elimination of AGEs. One of the important parts of this article would be to summarize the current AGEs inhibitors to gain insight into the anti-glycation mechanism of the skin and the development of promising natural products with anti-glycation effects.

## 1. Introduction to Advanced Glycation End Products (AGEs)

### 1.1. Introduction

Advanced glycation end products (AGEs) are complex heterogeneous molecules formed by the non-enzymatic reaction between reducing sugars (including fructose and glucose) and proteins, lipids, or nucleic acids with subsequent chemical rearrangements resulting in protein crosslinking, browning, and fluorescence [1]. AGEs can be crosslinked through side chains to form a substance of very high molecular weight, which is not easily degraded. Many cells have AGEs receptors on the surface, such as macrophages, mesangial cells, fibroblasts, and endothelial cells, through which their function can be affected. AGEs are ubiquitous in humans and accumulate widely in tissues with normal aging. They trigger inflammatory pathways by forming crosslinks, altering protein structure, or by binding to AGEs receptors. AGEs can lead to aging at high concentrations [2]. They are known to also be associated with age-related diseases, such as diabetes, atherosclerosis, renal failure (uremia) [3], kidney disease [4], age-related macular degeneration [5], aging phenotype, and Alzheimer’s disease [6]. Different aspects of AGEs are summarized in Figure 1.

### 1.2. Composition

AGEs are formed through four pathways: (1) the Maillard reaction, (2) sugars auto-oxidation pathway, (3) lipid peroxidation pathway, and (4) polyols pathway. Glucose is converted into fructose via the polyol pathway (based on aldo-keto reductase enzymes), which accelerates the production of AGEs [7]. AGEs precursors can also be formed by metal-catalyzed auto-oxidation of glucose (Wolff pathway) and lipid peroxidation (acetyl alcohol pathway) [8]. AGEs are complex and heterogeneous, and not all AGEs have been identified. Some of the unidentified compounds are derived from the complex mechanisms of AGEs formation and a wide variety of precursors, including Amadori rearrangement products, propionose (i.e., glyceraldehyde (GA)), and glycolytic intermediates [9]. More than a dozen AGEs have been detected in tissues and can be divided into three categories according to their biochemical properties: (1) crosslinked fluorescent AGEs, such as pentosidine, glyoxal-lysine dimer (GOLD), and crossline; (2) crosslinking non-fluorescent AGEs, such as methylglyoxal-lysine dimer (MOLD), alkyl formyl glycosyl pyrrole (AFGP) crosslinks, and arginine-lysine imidazole (ALI) crosslinks; (3) non-crosslinked AGEs, such as N(ε)-carboxymethyl lysine (CML), N(ε)-(1-carboxyethyl)lysine (CEL), MGO-derived hydroimidazolone (MG-H1), GO-derived hydroimidazolone (G-H1), and pyrraline [10]. Glycated hemoglobin (HbA1c) was the first observed endogenous early glycation product and was used as a suitable marker for average blood glucose levels [11]. The most prevalent AGEs in the human body, including the skin, is CML, which is formed by oxidative denaturation of Amadori products or the reaction of lysine with dicarbonyls (e.g., GO, MGO) formed during glycation. Pentosidine is also another AGEs, which is used as an important indicator for assessing glycation, and its formation and accumulation reflect a wider range of chemical modifications and damage to proteins, mostly produced through the reaction of pentose with lysine and arginine residues. Ribose produces AGEs faster than glucose and reacts with proteins to form Amadori compounds and oxidation to form MG-H1 [12].

Another classification of AGEs focuses on modified proteins, including AGE-1 (glucose-derived AGEs), AGE-2 (glyceraldehyde-derived AGEs), AGE-3 (glyoxal-derived AGEs), AGE-4 (methylglyoxal-derived AGEs), etc. [13]. Glycer-AGEs from GA, known as toxic AGEs (TAGE), are highly cytotoxic, and their accumulation can lead to various cellular disorders. TAGE leakage affects surrounding cells and increases serum TAGE levels, thereby promoting the onset and/or development of lifestyle-related diseases (LSRD) [14]. Typical AGEs derived from dicarbonyls and structures are shown in Figure 2. Given the heterogeneity of the formed AGEs, it is often difficult to quantify their formation completely. The most commonly used methods currently are combined with mass spectrometry. The immunoenzymatic method is also a popular method; CML, CEL, and pentosidine are often used as biomarkers for glycation processes. The fluorescence of AGEs is another marker. However, not all AGEs have fluorescence, and fluorescence-related AGEs can be used for fluorescence assays [15].

### 1.3. Sources

AGEs can be exogenously ingested (through food consumption) or endogenously produced and formed both intracellularly and extracellularly. Conditions such as diabetes mellitus would accelerate the metabolic processes of AGE formation [16]. The rate of AGE formation is thought to depend on carbohydrate precursors (intermediate glycation end products), concentrations of reactive oxygen species, and protein turnover [17]. Endogenous AGEs are formed through the Maillard reaction in three main stages: early, middle, and final. In the “early” stage, glucose reacts with free amino groups, including proteins, nucleic acids, and lipids. This will form unstable aldehyde amide compounds, the Schiff base, which are rearranged into Amadori products. In the “intermediate stage”, Amadori products are degraded into various highly reactive dicarbonyls (GO, MGO, 1-DG, 3-DG) as precursors of AGEs that react with free amino groups of proteins. At the “final stage”, these dicarbonyls react again with free amino groups, through oxidizing or non-oxidizing pathways, forming fluorescent, irreversible brownish compounds (often referred to as AGEs). The automatic oxidation of Amadori products resulting in AGE is called the Hodge pathway [18]. AGEs precursors can also be formed by metal-catalyzed self-oxidation of glucose (Wolff pathway) and lipid peroxidation (acetyl alcohol pathway), as well as degradation of corresponding Schiff bases and Amadori products. Free radicals are also involved in the formation of AGEs as reaction intermediates, and the mechanisms behind them include the self-oxidation of glucose in the presence of metal ions, producing ketoaldehydes and free radical superoxides; Amadori products automatically oxidize in the presence of intermediate metals and molecular oxygen, producing free radical superoxides [19]. The production of more reactive oxygen species (ROS) and reactive carbonyl groups (RCS) increases the number and type of AGEs. Endogenous AGEs form slowly in vivo and can be performed inside proteins to form high molecular weight (HMW) AGEs or between small molecules to form low molecular weight (LMW) AGEs [20]. Generally, only long-lived extracellular proteins can accumulate AGEs, which can also be formed on short-lived molecules or intracellular growth factors but are quickly eliminated due to a high tissue update rate [21].

Exogenous AGEs (mainly from food and tobacco) are important contributors to the human AGEs library, and they are structurally and functionally indistinguishable from endogenous AGEs [22]. The modern Western diet is full of heat-treated foods, which contribute to the intake of AGEs. Heat treatment accelerates the formation of AGEs in food [19] and results in a significant increase in circulating AGEs, including CEL, MGO, and CML [23]. Around 20% to 50% of the ingested CML appear to be excreted from the stool; a portion of the ingested AGEs are believed to be accumulated in the body. There is growing evidence that excessive consumption of AGEs can lead to inflammation and oxidative stress, which have an impact on many chronic disease states. AGEs diet restriction reduces chronic low-grade inflammation and can be used as a therapeutic strategy to alleviate chronic disease [24].

### 1.4. Mechanism of Injury

AGEs are involved in the vicious cycle of inflammation, production of ROS, and the amplification of AGEs [25], causing damage in four main ways. (1) Intracellular glycation—through protein modification, AGEs directly alter the structure of enzymes, proteins, lipids, and nucleic acids, thereby altering their properties and functions, interfering with cellular activity across the board and causing endoplasmic reticulum (ER) stress. Lysine and arginine residues of proteins are involved in enzyme activity sites, and modifications of these groups by AGEs can lead to enzyme inactivation. (2) AGEs can act as a catalytic site for free radical formation, exacerbate intracellular oxidative stress, and increase ROS production through a variety of mechanisms, such as reducing the activity of superoxide dismutase (SOD) and catalase, reducing glutathione storage, and activation of protein kinase C, etc., to accelerate the production and accumulation of AGEs. (3) AGEs bind directly or indirectly to AGE-specific receptors on various cell surfaces, activate NF-κB and mitosogen-activated protein (MAP) kinase signaling pathways, increase matrix metalloproteinase (MMP) production [26], and induce increased intracellular reactive oxygen species (ROS) [27] and inflammatory cytokine production [28]. (4) AGEs capture and crosslink macromolecules, which can alter their function. The crosslinking of elastin reduces viscoelasticity and causes the skin and vascular system to harden. The crosslinking of lens α-lens proteins leads to a decrease in lens transparency and increased light scattering [29].

Receptors that have an affinity for AGEs include AGEs receptor (RAGE), oligosaccharide transferase (OST)-48 or AGE-R1, 80K-H or AGE-R2, galactose lectin-3 or AGE-R3, and scavenger receptor (SR-A; SR-B: CD36, SR-BI; SR-E: LOX-1; Stab1; Stab2) [30]. Among them, the receptor for advanced glycation end products (RAGE) is the most widely studied. RAGE is a multiligand transmembrane receptor, which includes full-length RAGE, soluble RAGE (sRAGE), and endogenous secretory RAGE (esRAGE). sRAGE is the soluble extracellular domain after RAGE’s proteolysis on the cell surface, which protects cells from damage caused by intact RAGE receptor activation by competitively binding AGEs with RAGE [31]. esRAGE is an alternative splicing of pre-mRNA. The activation of RAGE receptors increases the solute reactive oxygen species via mitochondrial dysfunction and activation of NADPH (reduced nicotinamide adenine dinucleotide phosphate), resulting in oxidative stress, activation of nuclear factor kB (NF-κB), and expression of pro-inflammatory cytokines, such as TNF-α, IL-6, and IL-1. For example, systemic lupus erythematosus (SLE) patients are at risk of intensified glycation and exhibit a deficiency in sRAGE [32]. The mean serum AGE concentration in patients with multiple sclerosis (MS) was higher than in healthy controls, whereas the sRAGE concentration was lower [33] and was upregulated after treatment with interferon β-1a [34]. The treatment with sRAGE, targeting the RAGE, inhibited endothelial cell proliferation enhanced by recombinant psoriasin [35]. Not all AGEs have the same affinity for RAGE; MGO-derived AGE has a strong affinity for RAGE, and CML-modified proteins cannot bind to RAGE to activate the inflammatory signaling pathways. Another group of AGEs cell surface receptors that function in opposition to RAGE, called AGE-R1, AGE-R2, or AGE-R3, are involved in regulating endocytosis and clearance of AGEs [36]. Some studies show that AGE-R1 inhibits AGEs-induced cellular oxidative stress [37].

### 1.5. Reducing AGEs—Strategies to Improve Healthy Lifespan

The most direct way to reduce the accumulation of AGEs is to reduce the intake of sugar and exogenous AGEs. In addition, the body’s anti-glycation defense mechanisms play a key role in removing glycation products [38], where AGEs are degraded by enzymes and broken down by receptors before being eliminated by the kidneys [22]. The glyoxalase system, including glyoxalase I (GLO-1) and II (GLO-2) system, is the primary detoxification pathway for reactive dicarbonyls and is critical for the detoxification of AGEs [39], and its overexpression extends life [40]. The DJ-1/Park7 detoxification pathway has the ability to repair early glycated intermediates and glycated guanine residues in nucleic acids and nucleotides. Both the ubiquitin–proteasome system (UPS) and the autophagic lysosomal proteolytic system (ALPS) help remove AGEs, which can operate autonomously or cooperatively. The autophagy system may play a role in the degradation of polymer complexes, which cannot be removed by the proteasome [41]. Oxidized protein hydrolase (OPH), a serine protease present in human erythrocyte cytosol, which degrades glycated modified proteins, and many herbal extracts have the effect of enhancing OPH activity. AGER1 is an AGEs receptor that binds, endocytoses, and degrades AGEs, which can inhibit RAGE expression and negatively regulate any oxidative stress and inflammation induced by AGEs. Consuming AGEs depletes AGER1 in fat cells, leading to increased inflammation, oxidative stress, and insulin resistance. sRAGE levels are positively correlated with AGEs levels and can act as a competitive antagonist against RAGE. Exogenous administration of sRAGE captures and eliminates AGEs in the loop, thus preventing tissue damage caused by AGEs by acting as bait receptors. RAGE has been linked to many chronic diseases [42], particularly implicated in aging and inflammaging. Blocking the interaction of AGEs with RAGE can inhibit the production of oxidative stress and inflammatory processes in tissues. It is a new way to inhibit the process of advanced glycation [43]. There are also many therapeutic drugs that target AGEs, including synthetic compounds, natural products, foods, and vitamin supplements, which inhibit the formation of AGEs by scavenging free radicals produced during saccharification, trapping dicarbonyls, chelating metal ions, and destroying covalent crosslinking in AGEs.

## 2. The Hazards of Skin Glycation

The skin is mainly divided into three layers: epidermis, dermis, and subcutaneous tissue, and it is the organ with the largest contact area between the human body and the external environment. It not only protects the body from damage from the external environment and avoids the loss of water from the body but also has a certain cosmetic effect [44]. The aging of the skin is first manifested as the aging of cells. Studies have shown that with aging, the proliferation and vitality of skin fibroblasts are reduced, leading to a decrease in the secretion of elastin fibers, collagen fibers, reticular fibers, and extracellular matrix in the dermis layer of the skin. This will result in a deepening and lengthening of skin wrinkles, pigmentation, and other manifestations of aging. A distinctive feature of aging at the molecular level is the gradual accumulation of non-enzymatically modified proteins, i.e., glycation, which produces skin problems, such as wrinkles, pigmentation, and yellowing of the skin color [45].

### 2.1. The Harm of High Glucose to the Skin

Glycation is an aging reaction of naturally occurring sugars and dermal proteins [46], which begins in early life, develops clinical symptoms at around 30, and progressively accumulates in tissues and skin due to the glycated collagens that are difficult to be decomposed [47]. AGEs derived from natural sugars (such as glyceraldehyde-3-PO4, glucose-6-PO4, and fructose) are formed several times faster than AGEs derived from glucose. Thus, glucose is the main source of energy for mammalian cells, fueling glycolysis and the tricarboxylic acid (TCA) cycle [48]. High-sugar foods activate the reward system of hypothalamic regulation to promote the intake of more foods that are easily metabolized as glucose [49]. A correlation has been shown between a high-sugar diet and elevated sugar levels in the blood and skin, and a low-sugar diet can reduce skin sugar levels [47]. 

In addition, the correlation between high sugar levels and skin aging can be seen in diabetic patients, where one-third of this population has skin complications [50]. A prominent feature of aging human skin is the fragmentation of collagen fibers, which severely damages the structural integrity and mechanical properties of the skin. Elevated levels of MMP-1 and MMP-2, increased lysyl oxidase (LOX) expression, and higher crosslinked collagen in the dermis of diabetic skin lead to the accumulation of fragmented and crosslinked collagen, thereby impairing the structural integrity and mechanical properties of dermal collagen in diabetes [51]. Collagen crosslinking makes it impossible for them to easily repair [52], resulting in reduced skin elasticity and wrinkles [53]. Keratinocytes and fibroblasts are the main cells involved in wound healing, but due to the high glucose (HG) microenvironment in diabetics, the functional state of these cells is impaired, thereby accelerating cellular senescence [54].

Long-standing high glucose regulates different metabolic pathways, leading to glycotoxicity or hyperglycemic stress [55]. These metabolic pathways include polyol pathways, glycolytic pathways, hexosamine pathways, protein kinase C (PKC) activation, and the formation of AGEs [56]. These changes accelerate the production of ROS, increase the oxidative reaction of lipids, DNA, and proteins in various tissues [57], and ECM disorders [58]. High sugar also induces senescence in keratinocytes and fibroblasts [59], upregulates p 16, p 21, and p 53 gene expression [60], induces oxidative injury [61], apoptosis [62], activates transcription factor nuclear factor kappa B (NF-κB) [63], promotes secretion of TNF-α, IL-1β, IL-6, and IL-8 [64,65,66], and upregulates the expression of AGEs.

### 2.2. Advanced Glycaion End Products Induce Skin Aging

Over time, glycation in vivo causes skin AGEs to accumulate, resulting in wrinkles, loss of elasticity, dullness, and decreased function of skin, which is one of the main mechanisms of skin aging [67]. AGEs cause pathological changes in the skin through three processes. First, AGEs interact with their specific cell receptors, altering the levels of soluble signaling molecules, such as cytokines, hormones, and free radicals. Second, in the process of non-enzymatic glycation reaction, a large number of reactive oxygen radicals are released, creating a state of oxidative stress, leading to a significantly reduced level of glutathione, VitC, and VitE in the body. This causes synthetic disorders of collagen in skin tissues. Third, AGEs alter the physical and biological properties of the original extracellular matrix proteins, such as collagen. The most important concentrations of AGEs in the skin are (from the highest to the lowest concentrations) glucosepane, CML, pentosidine, and CEL [68]. Skin autofluorescence (SAF) has been shown to be a biomarker of cumulative skin AGEs [69], and measuring facial fluorescence intensity allows for an assessment of the skin glycation index [67]. SAF is also a powerful and independent predictor for cardiovascular disease and type 2 diabetes (T2D) [70]. Most compounds on the cosmetic market focus on blocking or reversing the initial saccharification reaction, i.e., the binding between proteins and sugars, reducing the formation of early saccharification Amadori products [71].

#### 2.2.1. Epidermis

The epidermis is the outermost layer of the skin, which has a protective function that prevents the penetration of pathogens and regulates the body’s water loss [72] and provides a natural “shield” against DNA damage [73]. Keratinocytes are the main cells in the human epidermis, which rapidly differentiate into four layers after proliferation: stratum corneum, stratum granulosum, stratum spinosum, and stratum basale. Protein turnover in the epidermis is much faster, but the accumulation of AGEs can still be observed for a short term before being replaced. mRAGE expression dominates the keratinocytes of healthy human epidermis and can monitor and respond to acute and cellular responses to maintain skin homeostasis [74]. The presence of RAGE suggests that AGE-mediated activation can have potentially negative consequences even for a short period of time. AGEs have been found to inhibit wound healing in diabetic patients by modulating the expression of MMP-9 in keratinocytes through the RAGE, ERK1/2, and p38 MAPK pathways [75] and inducing apoptosis and inhibiting normal cell growth by activating NF-κB [76]. 

Glyoxalase is detected in the epidermis and dermis, with GLO-1 located mainly in the basal layer of the epidermis, while GLO-2 being more prominent in the upper keratinocytes. The accumulation of AGEs can be counteracted by the enzymes GLO-1 and GLO-2 of the glyoxalase system, which works synergistically to detoxify the reactive precursors of AGEs. GLO-1 and GLO-2 are more abundantly expressed in older skin. Probably as a protective mechanism, the amounts of AGEs in the basal epidermal layer of the skin are lower. Photoexposure reduces GLO-2 production and thus promotes the accumulation of AGEs [77]. These results suggest that the glyoxalase system plays an important role in both chronological (intrinsic) aging and photoaging and acts as a defense system against skin aging [78]. The natural vitamin B_6_ analog pyridoxamine has been described as an anti-glycating agent, which not only quenches MGO but also increases GLO-1 activity. In addition, polyphenols, such as resveratrol and fisetine, can also upregulate GLO-1 expression [79]. 

#### 2.2.2. Dermis—Fibroblast

Fibroblasts are the main repair cells in the dermis but also the main cells that secrete collagen, and their normal proliferation and growth have great significance for maintaining the normal structure and physiological function of the skin. AGEs induce fibroblast senescence, matrix molecule proliferation (type I collagen, type III collagen, and type IV collagen), and metalloproteinase production (MMP1, MMP2, and MMP9) [80]. They also promote apoptosis, reduce hyaluronic acid (HA) synthesis, and reduce elastase-type matrix metalloproteinase (ET-MMP) activity. Finally, they regulate cell dysfunction by interacting with cell membranes and accelerating the leakage of lactate dehydrogenase (LDH) from cells [81]. AGEs modify intracellular molecules, including intermediate filament waveforms and proteasomes. The intermediate filament waveform is the main target of CML in human skin fibroblasts. DNA is also sensitive to glycation. GO causes DNA strand breaks, and MGO produces extensive DNA–protein crosslinking. RAGE levels have been found to increase over time, particularly in fibroblasts in the epidermal basal and upper dermis in elderly patients, and those interactions between AGE and RAGE lead to slower cell replication and induce a pro-inflammatory cascade [82]. On the other hand, fibroblasts may play an important role in skin AGEs degradation and photoaging skin AGEs accumulation. AGEs can be internalized by fibroblasts through receptor-mediated endocytosis and further degraded by lysosomal proteases or proteasomes. Among them, protease D plays a major role in the degradation of intracellular AGEs [83].

#### 2.2.3. Dermis—Extracellular matrix (ECM)

The extracellular matrix (ECM) is a complex, non-cellular network produced primarily by fibroblasts, including proteoglycans, hyaluronic acid, adhesion glycoproteins (fibronectin and laminin), and fibrin (collagen and elastin), as well as growth factors and cytokines [84]. They provide the mechanical strength and elastic resilience to the skin. ECM saccharification is manifested as increased skin hardness, decreased elasticity, activation of RAGE, and induction of fibroblast senescence and apoptosis [85]. AGEs can regulate the expression of ECM proteins and alter the expression and synthesis of the enzymes responsible for their degradation. AGEs have been reported to reduce ET-MMT activity in a dose-dependent manner and have a regulatory effect on matrix metalloproteinases (MMPs) [86]. The production of AGEs is primarily through non-enzymatic glycation of proteins in ECM. Longevity molecules are particularly susceptible to glycation, making collagen a common target for active compound modification due to its abundance and slow turnover rate. Glucosepane is the most abundant AGE that crosslinks the collagen of aging human skin. Due to its abundance, glucosepane is thought to play a major role in increasing skin stiffness and hardness. CML is one of the main AGEs in the skin and serves as an indicator of collagen glycation [87]. Fluorescent pentosidine is a recognized marker for general AGE accumulation, and the presence of pentosidine may indicate a significant increase in the level of glucoside and other AGEs [88]. In addition, glycated collagen induces CML expression in the dermis and epidermal compartments, resulting in an aging phenotype of poor stratification of the epidermal layer and keratinocyte cytoplasmic vacuolization [89]. If collagen is severely crosslinked, collagenase fails to degrade the modified collagen, causing AGEs to accumulate in the dermal skin [81].

### 2.3. UVA Induces Advanced Glycation End Products of the Skin

UVA exposure combined with dermal glycation are two catalysts for skin aging [90]. The accumulation of glycation products increases with age and is amplified by ultraviolet exposure [91]. AGEs produce superoxide anion radicals (O2^−^) and hydroxyl radicals (OH▪) after UVA irradiation, increase oxidative stress in the dermal matrix [92], damage human dermal fibroblasts, and accelerate the formation of the sugar oxidation products pentosidine and CML in actinic elastic tissues. Elastin crosslinked with AGEs cannot be degraded by elastase. UVA irradiation combined with AGEs enhances MMP1 and MMP3 mRNA expression, which induces protein expression of fibrin 1 and tropoelastin and reduces the expression of glyoxalase, which detoxifies harmful precursors of AGEs. Because glycated collagen and elastin are highly resistant to MMP degradation, a large accumulation of glycated proteins [93] leads to skin aging and elastic tissue proliferation [94]. In addition, keratinocytes secrete AGEs in response to UV irradiation, which stimulates melanin production through ERK and CREB signaling of RAGE, leading to skin pigmentation [95]. 

Figure 3 shows the effects of UV exposure combined with AGEs on the skin.

## 3. Inhibitors of AGEs

AGEs inhibitors are mainly divided into five categories: (1) carbonyl trapping agents that weaken carbonyl stress; (2) metal-ion chelators or scavenging free radicals, inhibiting sugar and lipid oxidation reaction; (3) crosslinking breakers that reverse AGEs crosslinking; (4) activating the anti-glycation system—many kinds of herbal extracts and natural compounds inhibit glycation by enhancing the anti-glycation system in the body; (5) RAGE antagonists. These include: anti-RAGE antibodies, sRAGE, and RAGE inhibitors; FPSZM1, a specific and potent chemical inhibitor of AGE receptor, which could improve diabetic nephropathy [96] and Aβ-mediated brain disorder [97]; Azeliragon, an oral small molecule antagonist of RAGE in Phase 3 development for mild cognitive impairment [98]. Small molecules are also in development to inhibit Diaphanous.1, the intracellular RAGE adaptor [99]. Inhibiting oxidative stress and inflammation in tissues by blocking the interaction of AGEs with RAGE is another new way to inhibit the process of late glycation. Figure 4 shows the action sites that inhibit the formation of AGEs in vivo.

### 3.1. Pre-Amadori Inhibitors

Aminoguanidine (AG) is an inhibitor of late glycation reactions in vitro found in clinical trials and is an excellent dicarbonyls scavenger that captures reactive carbonyl precursors, such as MGO, GO, and 3-DG. Amadori compounds are important intermediates for AGEs formation in vivo, and CML must be formed primarily by oxidative cleavage of Amadori’s Enediol intermediate between C_2_-C_3_ of the ligated sugar. AG was found to have no significant effect on the CML produced during the incubation of Amadori proteins. Therefore, AG is an important pre-Amadori inhibitor. AG is toxic at higher concentrations and has been forbidden in human clinical trials [100]. AG inhibits the development of diabetic complications in animal models of diabetes but does not inhibit the formation of late glycation end products of skin collagen in diabetic rats [101]. Benfotiamine, a synthetic thiamine precursor, activates the enzyme transketolase to accelerate the precursors of AGEs toward the pentose phosphate pathway, thereby reducing the production of AGEs [102].

### 3.2. Post-Amadori Inhibitors

Pyridoxamine (PM), one of the natural forms of vitamin B_6_, uniquely targets the post-Amadori pathway through metal-ion chelation and blocking oxidative degradation of Amadori intermediates [103]. Good post-Amadori inhibitor compounds should form stable metal-ion complexes with a higher equilibrium constant than the Amadori compound [104]. PM also has the ability to scavenge toxic carbonyl products from sugar and lipid degradation, inhibit reactive oxygen species [105,106], and increase the activation of the detoxifying enzyme GLO-1 [107]. The ilex paraguariensis (IP) extract is also a post-Amadori inhibitor due to its inhibition of the second stage of glycation reaction and conversion of free-radical-mediated Amadori products to AGEs [108].

### 3.3. Crosslinking Breaker

Thiazole salts are AGEs crosslinking breakers, such as OPB-9195 and ALT-711 (alagebrium). OPB-9195 inhibits AGE formation (particularly pentosidine and CML) through the chelation of metal ions and carbonyl trapping [109]. ALT-711 is the first compound in the thiazole class, which has been reported to break down established AGE-related cross-links. Another prototypic AGE cross-link breaker is N-phenacylthiazolium bromide (PTB) which break down protein cross-links by cleaving α-diketone structure. Similar effects have been observed with rosmarinic acid, tannins, and flavonoids [110]. There are also other potent AGEs destroyers, such as curcumin and ALT-946 [111].

### 3.4. Indirect AGEs Inhibitors

A small number of AGEs inhibitors play a role in the early stages of glycation by disturbing the initial binding between sugars and amino groups and indirectly reducing the formation of AGEs and ALEs. Since AGEs are mostly produced by non-enzymatic glycation of sugars and lipids, hypoglycemic and lipid-lowering drugs can inhibit the production of AGEs in vivo. For example, Atorvastatin (a lipid-lowering drug) inhibits the further formation of Schiff bases and AGEs by interfering with the initial binding between reducing sugars and amino groups [112]; Metformin is used to treat type II diabetes mellitus by inhibiting the production of reactive oxygen species by reducing the expression of the AGEs receptor (RAGE) [113] and capturing MG and other dicarbonyls produced during glycation. Buformin inhibits the formation of AGEs by trapping the carbonyl groups of ammonia and MGO and is a more effective inhibitor of AGEs formation than metformin [114]; Aspirin, or acetylsalicylic acid, inhibits the glycation process by acetylating the proteins’ free amino groups, thereby blocking the attachment of reducing sugars.

### 3.5. Natural AGEs Inhibitors

Synthetic AGE inhibitors have safety concerns and side effects, so natural products with lower toxicity are the most promising alternatives for developing natural medicines with anti-glycation activity. It has been reported that tea, herbal tea, vegetables, fruits [115], yogurt, and other foods have an inhibitory effect on the saccharification reaction. A large number of experiments in vitro and in vivo have shown that natural compounds have the potential to combat the formation and accumulation of AGEs, including phenols, oligosaccharides and polysaccharides, carotenoids (e.g., β-carotene), saponins [116], and unsaturated fatty acids.

Plants have long been used in traditional medicine techniques to treat various diseases and are also a source of new natural medicines discovery. Plant extracts have great anti-aging potential and are rich in a variety of active ingredients, which can inhibit the formation of AGEs by scavenging free radicals, capturing dicarbonyl carbon, etc. [117]. For example, C. ternatea flower extract (CTE) prevents protein glycation by trapping carbonyl groups and scavenging free radicals [118]. The polyphenolic components of peanut peel include gallocatechin, phenolic acids, and resveratrol, which reduce toxicity caused by AGEs and reduce the levels of reactive oxygen species and pro-inflammatory cytokines [119]. Citrus fruit extract significantly reduces the level of protein carbonyl compounds [120]. Akebia quinata fruit extracts (AQFE) can act as an anti-skin aging agent by preventing oxidative stress and other complications associated with AGEs formation [121]. Phenolic components of milk thistle flowers have anti-glycation activity in vitro and on human explants. Polyphenol-rich clove extract, due to its antioxidant properties, is able to inhibit the formation of AGEs and protein glycation [122]. The polyphenol compounds of hazelnut bark extract can reduce the formation of AGEs in vitro [123]. The hydrophobic extract of dunaliella salina, rich in colorless carotene phytoene and phytofluene, has anti-glycation and anti-inflammatory activity and helps reduce the signs of aging (wrinkles) [124]. Cinnamon is a traditional spice, which contains some phenolic components in its aqueous extracts, such as catchin, epicatechin, and procyanidin B2, which inhibit the formation of AGEs through antioxidants and direct capture of active carbonyl substances [125]. Black galangal extract inhibits the formation of fluorescent AGEs, pentosidine, CML, and intermediates 3-DG, GO, and MGO, and it acts on the decomposition of AGEs, thereby reducing the accumulation of AGEs in vivo [110]. Salvia officinalis L. methanol extract, including rosmarinic acid, resveratrol, quercetin, rutin, and luteolin-7-O-glucoside, exerts anti-glycation effects through antioxidation and inhibition of fluorescent substances and carbonyl groups [126]. Pomegranate fruit extract (PE), its phenolic constituents (punicalagin, ellagic acid, and gallic acid), and products of the degradation of ellagitannin (urolithin A and urolithin B) [127] all have effective anti-glycation activities [128].

### 3.6. Polyphenolic Compounds

As major and ubiquitous phytochemicals, including flavonoids, phenolic acids, alfalfa, and lignans, polyphenols exert AGEs inhibition through ROS inhibition, dicarbonyls capture (MGO and GO), and disruption of protein crosslinking [124,129]. Glycation and oxidative stress are closely related, with all steps of sugar oxidation producing oxygen radicals and ultimately resulting in the formation of AGEs. In addition, glycated proteins activate membrane receptors such as RAGE through AGEs and induce intracellular oxidative stress and pro-inflammatory states [130]. Therefore, compounds that scavenge free radicals can effectively inhibit glycation. For example, resveratrol (3,4′,5-Trihydroxystilbene) is a plant polyphenol that reduces oxidative stress [131] and inhibits AGEs-induced proliferation, collagen synthesis, and RAGE receptors [132]. Asiatic acid (AA), a pentacyclic triterpenoid, occurs naturally in many vegetables and fruits. AA pretreatment effectively protects HaCaT cells from subsequent AGE-BSA-induced oxidative and inflammatory stresses, exerting an anti-glycation effect [133]. The natural antioxidant “ellagic acid” (EA) exerts its inhibitory effect on AGEs in diabetic rats by inhibiting glycated ntermediates (including dicarbonyls) and interrupting the auto-oxidation pathway [134].

Flavonoids (flavones, flavanones, isoflavones, and flavonols) are the most common class of polyphenol compounds and have shown significant inhibitory effects on protein glycation and AGEs formation. These mechanisms may involve capturing reactive amino groups, so that they cannot react with glucose or scavenging carbonyl compounds, chelating with trace metal ions that catalyze glycation, scavenging hydroxyl radicals, and inhibiting oxidative degradation of various intermediates [135]. For example, garlic can inhibit protein glycation and dicarbonyls in vitro; quercetin is a phenolic compound found in garlic [136], which has a more effective anti-glycation effect than aminoguanidine [137]. Dietary antioxidants such as quercetin can prevent free radical toxicity [138]. Rutin (flavonoids) is found in fruits and vegetables, making it unable to react with glucose through mechanisms such as capturing reactive amino groups. All five metabolites formed after ingestion effectively inhibit the formation of CML [135]. Anthocyanins are the main flavonoids in blackcurrants that effectively prevent the formation of AGEs by capturing methylglyoxal [139].

Phenolic acids are secondary metabolites that are widely present in plants, including a large distribution of hydroxycinnamic acid (coumalic acid, caffeic acid, ferulic acid, coumarin) and hydroxybenzoic acid (Protocatechuic acid, gallic acid, hydrobenzoic acid, and ellagitannin). These metabolites are also found to have anti-aging potential. For example, ferulic acid inhibits the formation of fluorescent AGEs and CML and reduces fructosamine levels. This leads to the prevention of protein oxidation through the reduction in protein carbonyl formation and protein thiol modification [140]. Isoferulic acid (IFA) is a powerful antioxidant and has an effective inhibitory effect on protein glycation and sugar oxidation [141]. Cinnamic acid and its derivatives reduce the levels of fructosamines, the formation of CML, and the level of amyloid cross-β structures [142].

### 3.7. Other AGEs Inhibitors

Carnosine is a naturally occurring dipeptide (beta-alanyl-l-histidine), which hinders the formation of protein carbonyl groups and has the ability to chelate transition metal ions, prevent MG-induced glycation, and reduce sugar-induced crosslinking [143], leading to a significantly lower AGEs levels in the epidermis and reticular dermis of human skin explants [144]. Piperazine-2,5-dione reduces the number of late glycation end products accumulated in human dermal fibroblasts with age. Vitamin D therapy may help lower AGEs levels, significantly reduce NF-κB activation, and increase sRAGE levels [145]. Zinc has antioxidant, anti-inflammatory, and anti-apoptotic potential. Zinc deficiency may stimulate the formation of AGEs, while zinc supplementation may inhibit the formation of AGEs and protein carbonyl groups through a variety of signaling pathways and improve AGEs-induced apoptosis and oxidative stress [146].

## 4. Conclusions and Future Directions

The non-enzymatic glycation reaction between reducing sugars and proteins is common in the human body and is accumulated in many tissues and organs with aging. Typically for the skin, AGEs trigger the crosslinking of collagen and elastin in the dermal ECM. This results in a decrease in skin elasticity and an increase in stiffness, thereby promoting wrinkles. The accumulation of brown AGEs leads to facial yellowing, dulling, and hyperpigmentation. Skin aging (including intrinsic and extrinsic aging) is a complex process, and glycation plays an important role in this phenomenon.

The glycation reactions and the corresponding intermediates and final products have a great impact on health and beauty, but the current research on anti-glycation active ingredients is mostly limited to blocking a certain link in the glycation generation chain. The most commonly used method for assessing anti-glycation is the in vitro bovine serum albumin (BSA) glycation system [147,148]. This allows some of our common dietary products, such as yogurt, tea, and fruit, to exhibit some anti-glycation effect [149,150,151,152,153]. Obviously, these are not enough for people to experience significant anti-glycation effects. We believe that anti-glycation cannot be limited to preventing the formation of glycated products. It needs to be explored from the perspective of metabolism and energy utilization, and the formation and accumulation of AGEs should also be considered as a decline in metabolic capacity and metabolic imbalance/disorder. Therefore, how to evaluate the anti-glycation efficacy and the corresponding related methods need to be considered and established.

Secondly, as mentioned above, saccharification is a continuous series of non-enzymatic reactions involving many intermediates and multiple pathways. Therefore, anti-glycation ingredients may play a role at different stages of glycation. For example, some compounds directly bind to sugars or proteins, capture dicarbonyl carbons, or scavenge free radicals, etc., all of which can prevent the Maillard reaction process or side reactions, thereby exerting anti-glycation effects. A question that needs to be asked would be whether combining these with the reactive groups makes these compounds “harmless”. Whether the Maillard reaction can continue or produce other effects requires further in-depth follow-up research. 

In addition, the mechanism of saccharification needs to be further studied. For instance, we need to look more at the role of the extracellular matrix in skin anti-glycation. Studies have shown that AGEs induce the secretion of collagen in fibroblasts and co-localization of AGEs with elastin. This could be considered as a protective effect of the extracellular matrix in competition with receptors of AGEs. However, at the same time, glycation of collagen was also found to induce CML expression in the dermis and epidermal compartments [89]. Therefore, the role of ECM in skin anti-glycation needs to be further studied. For skin care, we should not only investigate the mechanism of skin glycation, but we should also investigate the efficacy of active substances during the entire anti-glycation process in vitro and in vivo. Specific factors, such as skin barrier and facial microbes, need to be considered to develop effective anti-glycation active ingredients.

## Figures and Tables

**Figure 1 nutrients-14-04588-f001:**
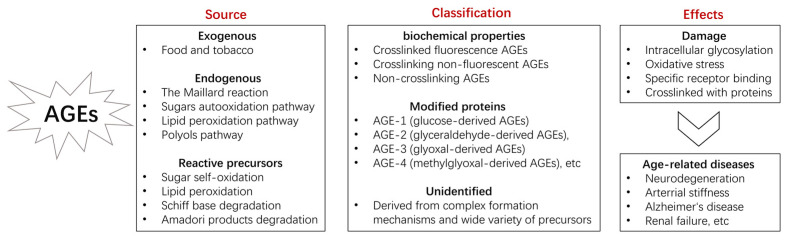
Introduction of AGEs.

**Figure 2 nutrients-14-04588-f002:**
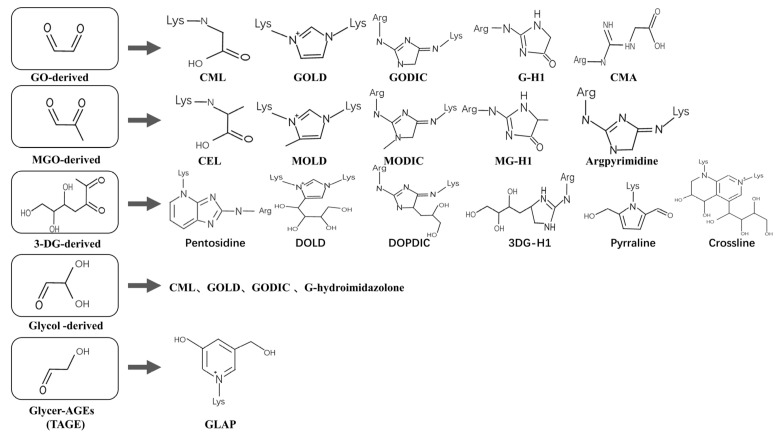
Typical AGEs and structures derived from dicarbonyls. Arg: Arg residue; Lys: lys residue; GO: glyoxal; MGO: methylglyoxal; Glycol: Glycolaldehyde; Glycer: Glyceraldehyde; DOLD: 3-deoxyglucosone-lysine dimer; CMA: N7–(carboxymethyl)arginine; 3-DG: 3-deoxyglucosone; GODIC: Glyoxal-derived lysine–arginine crosslinks; MODIC: methylglyoxal-derived lysine–arginine crosslinks; 3DG-H1: 3-deoxyglucosone-derived hydroimidazolone; GLAP: glyceraldehyde-derived pyridinium; DOPDIC: N6-{2-{[(4S)-4-ammonio-5-oxido-5-oxopentyl]amino}-5-[(2S,3R)-2,3,4-trihydroxybutyl]-3,5-dihydro-4H-imidazol-4-ylidene}-L-lysinate.

**Figure 3 nutrients-14-04588-f003:**
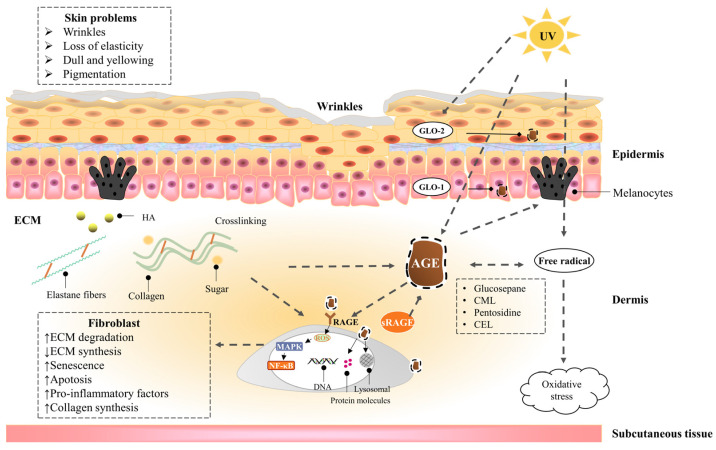
The effects of UV exposure combined with AGEs on the skin. A dotted line with an arrow indicates an induced effect; a dotted line with a diamond shape indicates a suppressive effect. AGEs in the skin are endogenously generated or exogenously ingested, including CML, CEL, pentosidine, and glucosepane, etc. Collagen is more likely to be glycated due to the slow turnover rate. On the one hand, AGEs act directly on cells, leading to a decrease in cell function by activating inflammatory signaling pathways and oxidative stress through cell surface receptors, as well as by modifying cell membranes and intracellular molecules, resulting in skin problems, such as dullness, pigmentation, and wrinkles. On the other hand, AGEs crosslink with collagen and elastin in ECM and promote the secretion of melanin, causing skin problems, such as macula and loss of elasticity. In addition, UV exposure can exacerbate skin glycation by promoting the generation of AGEs, exacerbate oxidative stress, and reduce epidermal GLO-2 production, leading to the accumulation of AGEs. AGEs can be degraded by proteases through receptor-mediated fibroblast endocytosis; the glyoxalase system can detoxify the reactive precursors of AGEs; sRAGE can competitively bind to AGEs with RAGE.

**Figure 4 nutrients-14-04588-f004:**
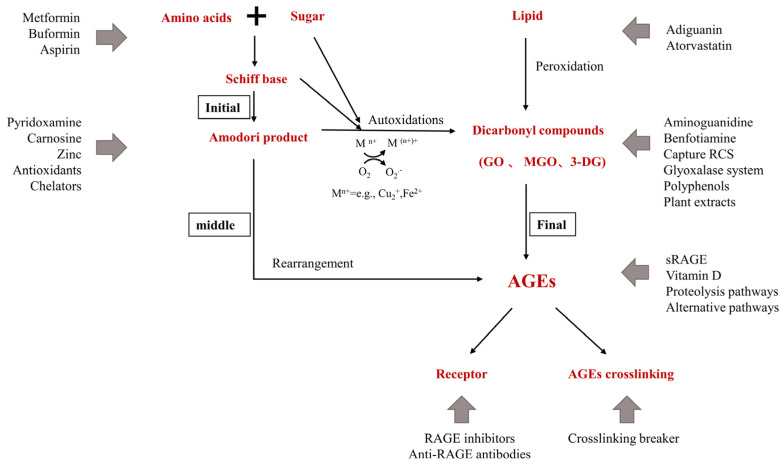
Action sites that inhibit the formation of AGEs in vivo. M^n+^ refers to transition metals. Endogenous AGEs are formed through the Maillard reaction in three main stages: early, middle, and final. The glyoxalase system include Glyoxalase I (GLO-1) and II (GLO-2) system; the proteolysis pathways include UPS and ALPS; alternative pathways include DJ-1/Park7 pathway, OPH, aldehyde dehydrogenases (ALDHs), aldo-keto reductases (AKRs), and acetoacetate degradation.

## Data Availability

Not applicable.

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
