# Peer review of "Research Advances on the Damage Mechanism of Skin Glycation and Related Inhibitors"

_nutrients, 2022, doi:10.3390/nu14214588_

Round 1

Reviewer 1 Report

Interesting manuscript, although it requires refinement.

There are no cited works in the field AGEs and their soluble receptor (sRAGE) in people suffering from multiple sclerosis, psoriasis  as well as from systemic lupus erythematosus (SLE) in the extensive literature.

 You would have to summarize the entire article in place of Conclusions as well as Future Directions section should be separate .

Detailed comments are in a file.

Reviewer 2 Report

In this review, by Wenge Zheng et al. entitled "Research advances on the damage mechanism of skin glycation 2 and related inhibitors", the authors focus their presentation on the damage mechanism of glucose and its glycation products on the skin by summarizing the biochemical characteristics, compositions as well as processes of the production and elimination of AGEs. One of the important parts of this review concern the current AGEs inhibitors to gain insight into the anti-glycation mechanism of the skin and the development of promising natural products with anti-glycation effects.

Please, find here my comments/recommendations:

1/ Abstract & Introduction. There is a mistake. This review is about glycation and not glycosylation. The last one is an enzymatic one while the first one is not. Glycation is a non-enzymatic reaction. Many corrections need to be done everywhere in the manuscript.

2/ Abstract & Introduction. Glycation is linked to age, diabetes, CV disorders… but also to renal failure (uremia)(See papers from T Miyata). See the paper from Semba et al. (J Gerontol, 2010) to present the main age-related diseases linked to glycation.

3/ Line 48. There is a mistake. The 3 glycation pathways are i) “classic” glycation, ii) polyols pathway and iii) glycooxidation. Schiff base and Amadori products belong to the “classic” glycation pathway and correspond to different steps of this same pathway. Glycoxidation has to be better defined.

4/ Line 189: RAGE has been described to be particularly implicated in aging and inflammaging.

5/ Line 253. SAF in an excellent maker of cardio-vascular risk. Is there any correlation between skin aging and cardio-vascular risk?

6/ Line 360. Concerning RAGE inhibitors, please introduce small molecules like FPSZM1 and Azeliragon. The last one has been tested in phase 3 clinical trial in the frame of Mild Cognitive Impairment. Small molecules are also in development to inhibit Diaphanous.1, the intracellular RAGE adaptor.

7/ Paragraph-Line 382. Benfotiamine must be added.

8/ A specific paragraph concerning glyoxalase.1 and its anti-glycooxidation effect must be added

9/ Paragraph-Line 415. Urolitine A is described as natural anti-glycation product like curcumin…

10/ Paragraph-Line 466: plant extracts must be included in the Naturals AGE inhibitors (Paragraph-Line 415
